# Diagnosing Juvenile Huntington’s Disease: An Explorative Study among Caregivers of Affected Children

**DOI:** 10.3390/brainsci10030155

**Published:** 2020-03-07

**Authors:** Mayke Oosterloo, Emilia K. Bijlsma, Christine de Die-Smulders, Raymund A. C. Roos

**Affiliations:** 1Department of Neurology, Maastricht University Medical Center, 6202 AZ Maastricht, The Netherlands; 2Department of Neurology, Leiden University Medical Center, 2333 ZA Leiden, The Netherlands; R.A.C.Roos@lumc.nl; 3Department of Clinical Genetics, Leiden University Medical Center, 2333 ZA Leiden, The Netherlands; E.K.Bijlsma@lumc.nl; 4Department of Clinical Genetics, Maastricht University Medical Center, 6202 AZ Maastricht, The Netherlands; c.dedie@mumc.nl; 5GROW Research Institute for Oncology and Developmental Biology, Maastricht University, 6200 MD Maastricht, The Netherlands

**Keywords:** juvenile Huntington’s disease, pediatric Huntington’s disease, early-onset Huntington’s disease, personal experiences, caregivers

## Abstract

*Objective:* To investigate the reasons for the diagnostic delay of juvenile Huntington’s disease patients in the Netherlands. *Methods*: This study uses interpretative phenomenological analysis. Eligible participants were parents and caregivers of juvenile Huntington’s disease patients. *Results*: Eight parents were interviewed, who consulted up to four health care professionals. The diagnostic process lasted three to ten years. Parents believe that careful listening and follow-up would have improved the diagnostic process. Although they believe an earlier diagnosis would have benefited their child’s wellbeing, they felt they would not have been able to cope with more grief at that time. *Conclusion*: The delay in diagnosis is caused by the lack of knowledge among health care professionals on the one hand, and the resistance of the parent on the other. For professionals, the advice is to personalize their advice in which a conscious doctor’s delay is acceptable or even useful.

## 1. Introduction

Huntington’s disease (HD) is an autosomal dominant neurodegenerative disease characterized by unwanted movements, psychiatric disorders, and cognitive deterioration. HD results from an unstable and expanded Cytosine Adenine Guanine (CAG) trinucleotide repeat in the Huntingtin (*HTT)* gene on chromosome 4 [1]. A CAG repeat size of 36 or more is invariably associated with HD. Most patients develop symptoms and signs in adulthood, with a mean onset of 40 years of age.

The juvenile form of Huntington’s disease (JHD) is rare. It is defined as HD with an onset of <21 years of age [2]. The Juvenile HD Working Group of the European Huntington’s Disease Network (EHDN) recently redefined JHD as pediatric HD with an age of onset ≤18 years [3]. However, as our study was conducted before this redefinition, we still use the old definition. JHD contributes about 5.4% of all HD cases, with the percentage ranging from 1%–15% in several series [4,5]. There is an inverse correlation between the length of the CAG repeat and the age of onset. The longer the CAG repeat, the earlier the disease-associated symptoms start [6,7].

The clinical presentation in children differs from that in adults. The most prevalent clinical features at the presentation of JHD are cognitive impairment and behavioral changes [8,9]. Common presenting motor features are rigidity, gait disorder, and oral motor dysfunction [2,4,10,11]. Chorea is uncommon in children with HD but becomes manifest in the second decade. Clinical features in the first decade are defined as two or more of the following features: declining school performance, seizures, oral motor dysfunction, rigidity and/or gait disorder in combination with a parent with (pre)symptomatic HD or a family history of HD [4]. The clinical features of individuals in the second decade (adolescent) are less well defined in the literature but are more comparable to the adult manifestation. Disease progression in JHD is likely to be faster compared to normal onset [12]. The variable and non-specific clinical presentation, such as declining school performances and behavioral disturbance, may be confused with disorders such as autism spectrum disorders or attention deficit hyperactivity disorder (ADHD) or with the effects of disrupted social and home environments in HD families. This significantly increases the chance of misdiagnosis and/or diagnostic delay. JHD is rare and probably less well recognized than usual-onset HD (30–50 years) by health professionals without specific knowledge of the disease. One study shows a mean delay of 9 years before diagnosis and another shows delays ranging from 0 to 6 years [9,11].

The psychosocial impact and personal experience of parenting a child with JHD have been described in earlier studies [13,14,15,16]. They describe the denial of parents at first, but also the awareness something is wrong with their child. Furthermore, these studies highlight the positive and negative experiences of the parents regarding the support they receive from family, friends, and professional caregivers.

Our aim is to investigate the diagnostic process of JHD patients in the Netherlands. We focus on the diagnostic timeline, the experiences of the parents or caregivers during the diagnostic process, and the role of the different health care providers. In this way, we hope to gain insight into the reason for the probable diagnostic delay and how to improve the diagnostic path.

## 2. Methods

This study employed in-depth semi-structured interviews and interpretative phenomenological analysis (IPA), a well-established experiential approach in health and clinical psychology [17]. IPA’s focus on the in-depth examination of the psychological process and descriptions of how individuals deal with life-transforming, or life-threatening events, conditions, or events.

### 2.1. Participants

Eligible participants were parents and/or caregivers of (living or already deceased) JHD patients in the Netherlands. Participants were recruited through a call published in the magazine of the Dutch Huntington’s disease association and through HD specialists. The patients themselves were not interviewed.

In this study, JHD was defined as the onset of symptoms and signs of HD before the age of 21 years [4]. Confirmation of the clinical diagnosis by molecular testing was not necessary for inclusion. If the patient was older than 20 years of age at the time of the interview, the caregivers could still participate.

All subjects gave their informed consent for inclusion before they participated in the study. The study was conducted in accordance with the Declaration of Helsinki, and the protocol was approved by the Ethics Committee of Leiden University Medical Center, The Netherlands (P17.025). Participants could also consent to a semi-structured interview with their child’s general physician (GP). Information retrieved from the GP was used to complete the timeline of the diagnostic process.

### 2.2. Data Collection

The interviews took place at the participants’ homes or the out-patient clinic and were conducted by the first author and audio-recorded. All interviews started with the opportunity for participants to ask questions about the aim of the study. The interviews were semi-structured, covering general issues regarding JHD (age of onset, first symptoms), the recognition of JHD symptoms by parents and health professionals, the number of consulted health professionals, and their expertise in HD. Furthermore, questions about the quality and quantity of support from health care professionals and the process of diagnosis were included. The following two questions, directly referring to the process of diagnosis, were always asked:

Which elements of the diagnostic process should be improved?

Should the diagnosis of JHD have been made earlier, and if so, would you and your son/daughter have benefited from it?

### 2.3. Analysis

Consistent with standard qualitative research techniques, the interviews were based on a topic list, which evolved as the interviews progressed through an iterative process to ensure the questions captured all relevant emerging themes.

The interviews were transcribed ad verbatim and analyzed with IPA. First, several close readings of the transcripts were made, and points of interest and significance noted and coded. Second, these initial comments were used to document emerging themes, aiming at capturing the essential quality of the participants’ experiences. Third, a list of themes was compiled and connections were made between the themes. The connected themes were grouped and labeled (superordinate theme). The superordinate themes for each interview were then compared, producing a table of comparative themes. 

## 3. Results

Eight parents/caregivers gave informed consent for an interview. The interviews lasted approximately 60 min each. Two parents contacted the researcher after reading the announcement in the magazine of the Huntington’s disease association. Five parents were approached by the treating HD specialist, and one parent was informed by another parent within the JHD community and contacted the researcher themselves. Six individual parents/caregivers and two couples were interviewed. Four of them also gave informed consent for an interview with their child’s GP. All identifying information has been changed to protect the privacy of the participating families. Table 1 presents the clinical characteristics of the children. The cultural backgrounds of the parents were alike. Seven participants had one affected child. One had two affected children. The mean age at diagnosis was 16 years, with a range of 9 to 24 (Figure 1). The mean repeat size was 62, with a range from 49 to 83. Individual repeat sizes cannot be provided since parents gave no informed consent for this. In six cases the father was the affected parent and, in three cases, the mother. All but one of the affected parents were deceased at the time of the interview. Six of the nine children were still alive. Six children had or still received HD expert care, three from an HD expert neurologist and three in an HD nursing home (Table 2).

Six superordinate themes emerged from the analysis. They describe the parents’ perceptions of the problems associated with the JHD diagnostic process. The six themes are (1) awareness (“something is wrong”), (2) the role of the different health care providers during the diagnostic process, (3) experiences after the diagnosis, (4) the need for support, (5) which elements of the diagnostic process should be improved, and (6) what if the diagnosis had been made earlier? These themes will be highlighted below by means of quotes from the interviewed parents/caregivers.

### 3.1. Awareness: “Something Is Wrong”

All parents noticed something was wrong with their child when, in retrospect, the first symptoms of the disease presented. The first symptoms varied from decreasing school performance, behavioral or motor symptoms, and sometimes a combination of these. Some initially thought the situation at home with the affected parent or the recent decease of a parent from HD was the cause of the behavior problems.

“My husband was still alive, but he was ill and I thought it’s the situation at home. Because there was so much going on at home. I thought it was the stress because dad is at home and also has all these weird complaints.” (Anna) 

“By the end of elementary school, his grades were so poor. But at that time I thought it was because of the huge problems at home with my husband.” (Clara)

These quotes highlight the disruption of the home environment with an ill parent as well as behavioral problems of this parent being seen as a cause of the behavioral changes in children from HD families. Also, puberty was mentioned as the initial presumed cause of the change in the behavior in older children.

“He was aggressive and angry. I thought it was some kind of macho behavior and wasn’t quite sure if it had anything to do with HD. We thought it was puberty. But then I started reading about HD and I knew it was Huntington’s.” (Ellen)

Ellen soon realized the behavioral changes were more likely to be caused by HD rather than puberty. Just like Francine, she is a stepmother. Their husbands both took a lot longer to realize something was wrong. 

“It was the first thing I thought. I thought this isn’t right. A normal student has the same grades throughout all their school years. Her father and grandmother thought it would be alright and that she was just emotional because her mother had just died.” (Francine)

Just like the other parents, Francine’s husband believed the death of his daughter’s mother was the cause of emotional disruption rather than the start of JHD. 

All parents indicated they thought the child’s behavior was likely caused by the illness of the affected parent, regardless if they were still alive or already deceased. However, the stepmothers, Francine and Ellen, were convinced that it was symptoms and signs of HD.

### 3.2. The Role of the Different Health Care Professionals during the Diagnostic Process

#### 3.2.1. Awareness of the Caregiver

After the initial search for explanations, the parents gradually started realizing the signs were probably associated with HD. None of them formally knew about the juvenile onset of HD at the time. They usually consulted their GP for advice.

“Looking back, I think I knew she had Huntington’s disease. I just didn’t want to hear it, and when a GP tells you it isn’t HD, that’s simply what you want to hear. When the institute said it was autism, I thought that’s just fine, even though I knew it wasn’t. It’s a very strange way to fool yourself, but you do it anyway.” (Anna)

“We went to see our GP, who told us to buy some good shoes, start some therapies. So there was about 2 to 3 years between our first visit to the GP and the neurologist.” (Geraldine)

When the thought of HD first crossed their minds, the parents felt anxious and unsure of what to do. They relied on the judgment of the health caregiver. Although most of the parents believed the physician misjudged the situation, they also admit they ignored their own suspicions that it could be JHD.

#### 3.2.2. Positive Support

The parents did experience helpful support from health care professionals in their search for an answer to their child’s problems. They mentioned health care professionals who took the problems seriously and/or who offered help and support, even though there was no diagnosis or a diagnosis other than JHD.

“Even though they didn’t make a diagnosis, that’s different than the feeling of being supported. Sometimes doctors simply don’t know what a patient is suffering from, but they do everything in their power to support you, psychologically as well as in other areas.” (Anna)

“We started therapy for autism. I asked one of the physiotherapists if he knew what Huntington’s disease was. He said he did, and he conducted several tests and put them on video to show a neurologist. After the summer holiday, he came back and said I was right, it was Huntington’s disease.” (Debby)

#### 3.2.3. Failure of Support

Three children were first diagnosed as having autism or ADHD by several health care professionals, ranging from neurologists to psychologists. 

“I remember that psychologist…Eventually, they had to make a diagnosis. She sat there with that green book on her lap, turning the pages back and forth. Finally, she said, well, we’ll go with ADHD. I don’t believe she was convinced at all.” (Debby)

“He started having behavioral problems and we went to see an HD expert neurologist. We already knew he had a 50% chance of HD. But the neurologist ran some tests and said he thought it was a form of Attention Deficit Disorder. He really didn’t see any signs of HD. We were told to come back if we started seeing any symptoms of HD.” (Geraldine)

The children presented with behavioral changes, such as aggressive behavior, hyperactivity, or obsessive behavior. This led to misdiagnoses such as ADHD and autism. Professional help clearly failed in the period before diagnoses when parents were starting to experience problems with their child. The follow-up period was too short or was lacking entirely. In four cases, the health professionals denied there was anything wrong at all. The parents had a clear need for support from a health care professional on a regular basis. 

### 3.3. Experiences after the Diagnosis

After the diagnosis had been made, most parents and their children received the support they needed. Three of the children received or are now receiving follow-up care from a neurologist with HD expertise. These children were 10 years or younger at the onset of HD. These parents are satisfied with the care provided. 

Geraldine’s two children received follow-up care from a neurologist with expertise in movement disorders, and she was very satisfied with the support they received.

“We went there twice a year. Our son liked him because I liked him (the neurologist with HD expertise). And there was a special team just for him. That was fantastic. They did so much. Incredible, yes.” (Beatrice) 

Geraldine and Harry received long-term support from a social worker from one of the clinical genetic departments in the Netherlands. They were also very satisfied with the support from the social worker.

“I must say she was fantastic (social worker). She really took care of us. My daughter bonded with her. She guided us after the diagnosis. And she was the only one my son would speak to. She deserves a huge compliment.” (Geraldine)

“She gave me advice. She gave me her mobile number. I once called her on a Sunday, and she rang me back within seconds. I could always tell her my story. Sometimes people complain about the care they get, but hats off for the care we got.” (Harry)

It is comforting for parents to receive care from a caregiver who has knowledge of the disease. They feel their problems are taken seriously, and the caregiver is able to help with some of their issues. It is crucial for them that the caregiver knows the course of the disease and is able to support them in the long-term.

### 3.4. The Need for Support

It was clear to Clara, Eric, and Harry that their child suffered from JHD and they did not feel the need for a formal clinical diagnosis. Their past experiences with caregivers regarding HD were either somewhat negative or they felt caregivers were not able to do anything that would contribute to their child’s well-being. 

“It was so obvious for us that it was HD, we didn’t do anything with it. We went to our GP once, and he had to look it up in a book. And even neurologists don’t know what to do with it. They say they know what HD is, but they don’t.” (Ellen and Eric)

“Well, how shall I put this? I don’t know if I’m allowed to say this, but there’s nothing a neurologist can do for him.” (Harry)

It was clear to Clara that something was wrong or that their child probably suffered from HD, but apart from visits to their GP, she did not feel the need for a referral. She assumed diagnosis was made by DNA testing and “as that is not allowed before the age of 18”, there was no use in visiting a specialist.

“In that period, I was aware there was something going on, and then I thought, what am I going to do about it? Because we can’t get him tested; he needs to be 18 for that. In fact, nobody made the diagnosis, but for us, it was clear.” (Clara)

Anna and Debby felt the need for a diagnosis, but no need for extra support from health care professionals when their child became ill. They explained that the problems at home with an ill or deceased spouse were so huge that there was no place for any kind of support at that time. However, in a later stage, they felt somewhat better and reported they would have liked some support.

“I didn’t feel the need. My husband had just died, so there was so much else to deal with at that moment. Looking back, I’d just had time to get back on my feet again when the next thing bombarded us.” (Anna)

Clara and Geraldine felt the need for support, but their children did not want medical care or HD expert care.

“The problem was she didn’t want HD expert care. They offered us a case manager who came a little too late after we’d already figured it out. We arranged 95% of the care ourselves because she wouldn’t let us involve anyone else.” (Geraldine)

As described before, many HD patients do not feel the need for support from caregivers or expert centers. However, the family can feel abandoned as they generally do feel the need for support. In their opinion, they could have received support or help even if their child did not want it.

### 3.5. Which Elements of the Diagnostic Process should be Improved?

Most parents said they felt there was no-one to listen to them and take them seriously. They would have also liked the health professionals to be upfront with them. Four parents visited several health care professionals who offered no follow-up visits, which they clearly would have appreciated.

“I wanted to hear it was nonsense, and he (the GP) told me it was. On the one hand, I was glad, but on the other, I think he should have done something with it, he should have kept an eye on it. Asked us to come in for a check-up a few months later to see how things were going.” (Anna)

Even after the diagnosis, some parents did not receive the right support until a later stage of the disease when their child had already been admitted to a nursing home. Clara’s son did not want support. However, she believes the health care professionals should have provided more assistance and listened to the parents.

“Well, look at when things get out of hand, and there are no care providers there. It’s clear they (patients) don’t want them there. They don’t want others to see what happens. Take the parents apart, ask how they experience things, how they see things. And then afterward decide how to move on.” (Clara)

Parents with an adolescent child with JHD often say their child does not want support from caregivers. This makes it difficult for parents to find long-term care and support. Denial of the disease and problems resulting from the disease are a huge problem in patients with HD. It sometimes takes a lot of patience to persuade HD patients they need long-term care.

Debby was offered support, but at that time, just after the diagnosis was made, it was too much for her as well. 

“We contacted a social worker from an HD expertise nursing home, but because we had already figured things out for ourselves, she couldn’t add much. However, it would have been good to have a case manager. And for you to be offered a case manager several times instead of once. If they offer this in a very hectic period, it might be too much.” (Debby)

Again, parents said they would have appreciated follow-up in time, even though it was not initially wanted or needed.

### 3.6. What if the Diagnosis of JHD Had Been Made Earlier?

Parents whose children received a diagnosis other than JHD felt they would have been able to treat their children differently with an earlier correct diagnosis. They would have paid more attention and would have had more patience with their child if they had known what was going on.

“I think he would have benefited from it. My husband had just died. Things just kept piling up and I paid less attention and certainly had less patience.” (Debby)

“When you know something’s wrong with your child, you approach it in a different way. I sometimes thought she was just lazy or didn’t feel like doing things. I think it would have made a difference in how I approached her. I wouldn’t have been so strict.” (Anna)

Furthermore, if the diagnosis had been made earlier, proper medication would have been provided in cases of aggression or other behavioral problems. 

“I think that if he’d had medication to suppress the aggression. That would have made life easier for him.” (Clara)

Francine believes her stepdaughter would not have been so alone. Before the diagnosis, she went to a regular school. However, because of her changing behavior, she did not have many friends and was isolated.

“Maybe she wouldn’t have been so alone and isolated from other children. She was so alone, that was incredibly sad. Of course, I don’t know how she experienced it.” (Francine)

All the interviewed parents believed an earlier diagnosis would have benefited their children. Especially in the different way they, as parents, would have approached their children. However, as mentioned before, some of the parents had difficulty acknowledging their child might have JHD. 

“On the one side, it would have been better if the GP had taken the suspicions of the psychotherapist seriously. He could have sent her to a neurologist. That would have been a more logical decision than telling me not to worry. I suppose it’s difficult for doctors to say or… it’s logical really, in case of such a terrible disease and knowing a child has a 50% chance, you would think they’d do something if there’s any kind of neurological issue. On the other side, I don’t know if I would have been able to handle it at the time.” (Anna)

An earlier diagnosis may not always have contributed to their well-being as parents. They felt they would not have been able to cope with any more grief or problems than they already had at that time.

## 4. Discussion

Our study describes the experiences of parents who have cared for or are still caring for a child with JHD, in particular, in regard to the diagnostic process of JHD. 

The parents described denial and, afterwards, being aware something was wrong with their child. This denial and awareness are clear phenomena mentioned in previous studies on JHD [13,14,15,16]. The most commonly perceived reasons for delayed diagnosis reported in an Australian survey among parents of children living with rare diseases was the lack of knowledge among health professionals [18]. In line with this, we found that the unawareness or lack of knowledge of this serious neurological disease among health care professionals led to a delay in diagnosis. Parents believe that careful listening, attention, and clinical follow up would have improved the diagnostic process.

However, we are the first to report that, for some parents, a period of denial was helpful. They indicated that if the diagnosis JHD had been made shortly after the presentation of the first symptoms, this would have been too much for them. In a way, growing slowly towards the realization of their child being affected by JHD was helpful. 

The fact that some of them were recently confronted with a sick or recently deceased spouse contributed to the denial. They would not have been able to manage an earlier diagnosis. In fact, the delay gave them the opportunity to process and face the problems with ill family members consecutively. Some parents visited a neurologist with expertise in HD and would have had the opportunity to discuss their suspicions with the neurologist. They chose not to do so.

So, it seems that shortening the diagnostic delay will not be helpful for every parent. For some children, the mixed feelings of their parents had probably added to a delay in diagnosis. The parents believed their child probably would have benefited from an earlier diagnosis as personalized care and pharmaceutical treatment could have been started earlier, and they would have been more patient with their child.

All children with JHD either received or were offered multidisciplinary care after the diagnosis. However, some parents turned down the offer of help because they either had so many problems that any form of support would have been too much, or their past experience with their affected spouse led them to believe this support would not be of much value. 

It seems difficult for clinicians with little or no experience with HD to diagnose JHD. However, HD specialists are still faced with the dilemma if non-specific signs symptoms are caused by JHD or a number of other explanations for the behavioral problems. Therefore, informing the HD community about the existence of JHD could help make parents more aware of the possible signs of JHD. This gives parents the opportunity to gain information if they wish to have it. Better information would also improve their understanding of the condition and of what to expect in the future, and it would probably help them better manage the challenges they face [15].

For clinicians specialized in HD, we recommend a careful follow-up if children from HD families present with non-specific symptoms such as decreased school performance, hyperactivity, and/or behavioral changes [4]. Children from HD families who present with such symptoms should be evaluated longitudinally, at least on an annual basis. Furthermore, we recommend being upfront about whether symptoms and signs might be due to JHD or not. On the other hand, we believe it is important to judge whether parents are able to cope with the diagnosis. Also, it is good to be aware that parents appreciate it when help and support are offered more than once, as the need for support can change during the course of follow-up.

We conclude that the delay in diagnosis is caused by the lack of knowledge among health care professionals on the one hand and the resistance of the parent on the other. In our opinion, this is a new and important finding that has not been described before. For health care professionals, diagnosing JHD is walking a tightrope. As for some parents, an earlier diagnosis would be too distressing; it is important to check whether parents are ready for the information. If not, keep in contact and try again at a later appointment. We hope these findings will be helpful for clinicians, caregivers, and the HD community in contributing to the well-being of these children and their parents. For professionals, our advice is to personalize their advice, in which a conscious doctor’s delay is acceptable or even useful.

## Figures and Tables

**Figure 1 brainsci-10-00155-f001:**
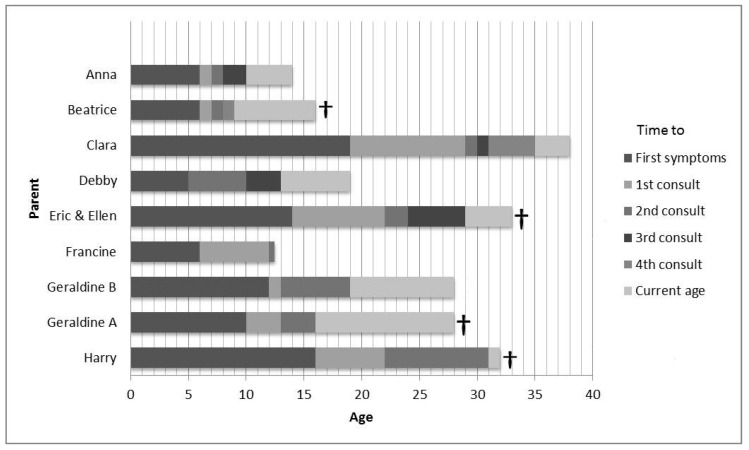
Diagnostic process timeline. † = death.

**Table 1 brainsci-10-00155-t001:** Clinical characteristics of juvenile Huntington’s disease (JHD) patients.

Parents	Affected Child	Affected Parent	Age of First Symptoms (y)	Present age (y)	Age of Diagnosis (y)	DNA	MDC	HD Expert Care	Diagnosis
Anna	daughter	F^†^	6–7	14	10	yes	yes	Yes	HD expert neurologist
Beatrice	Son	F	6	16^†^	9	no	yes	Yes	HD expert Neurologist
Clara	Son	F^†^	12–19	38	28	yes	no	Yes	clinical genetics
Debby	Son	F^†^	5	18	13	yes	yes	No	child neurologist
Eric & Ellen	Son	M^†^	14–15	33 ^†^*	24	yes	no	Yes	clinical genetics
Francine	stepdaughter	M^†^	6	15	10	yes	yes	Yes	HD expert Neurologist
Geraldine A	daughter	F^†^	10	28 ^†^*	16	yes	yes	Yes	MD neurologist
Geraldine B	Son	F^†^	12	28	19	yes	no	No	MD neurologist
Harry	Son	M^†^	16	31^†^	22	yes	yes	No	clinical genetics

F = father; M = mother; † death; * euthanasia; MDC = multidisciplinary care; HD = Huntington’s disease; MD = movement disorders.

**Table 2 brainsci-10-00155-t002:** Diagnostic process by consult.

Parent	1^st^ Consult	2^nd^ Consult	3^rd^ Consult	4^th^ Consult
**Anna**	Psychologist neurological disorder	Psychologist ASD	HD expert HD	
**Beatrice**	physiotherapist/school doctor neurological disorder	MHS ASD	neurologist/pediatrician spasticity	HD expert HD
**Clara**	clinical genetics HD	MHS	mental hospital	HD nursing home
**Debby**	psychologist ADHD	autism expert team ASD	Neurologist HD	
**Eric**	clinical genetics HD	neurologist	HD nursing home	
**Francine**	MHS Unknown	HD expert HD		
**Geraldine A**	Neurologist HD	HD expert HD		
**Geraldine B**	HD expert ADD	clinical genetics/neurologist HD		
**Harry**	clinical genetics unknown	assisted living	HD nursing home HD	

All referrals took place after a visit to the general practitioner. All HD experts are neurologists. Abbrevations: HD = Huntington’s disease; MHS = mental health services; ASD = Autism Spectrum Disorder

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
