# Peer review of "Diagnosing Juvenile Huntington’s Disease: An Explorative Study among Caregivers of Affected Children"

_brainsci, 2020, doi:10.3390/brainsci10030155_

Round 1
Reviewer 1 Report
Qualitative paper with information about delay to diagnosis. Eight family carers were interviewed. It would have been better to have had the CAG repeat number and some comments on why there may have been misdiagnosis of ADHD for example. Also, was there any difference in response from carers who had the spouse/other parent affected by HD still alive. It's also interesting that if there was a known affected parent, there was still diagnostic delay - is there any literature on this?
Author Response
Thank you very much for reviewing our paper on juvenile onset of Huntington's Disease. Below you will find the answers to your comments and questions.
Comment 1: It would have been better to have had the CAG repeat number
Answer 1: we have not the informed consent of all the parents to publish the individual repeat size. However, from most of them we have the repeat size. Therefore, we provide the mean repeat size of the patients. In the main text I added the following on page 3, line 120-121: The mean repeat size was 62 with a range from 49 to 83. Individual repeat sizes cannot be provided, since parents gave no informed consent for this.
Comment 2: It would have been better to have had some comments on why there may have been misdiagnosis of ADHD for example
Answer 2: To clarify the misdiagnosis I added the following line: The children presented with behavioral changes, such as aggressive behavior, hyperactivity or obsessive behavior. This lead to misdiagnosis such as ADHD and autism. Professional help clearly failed in the period before diagnoses when parents were starting to experience problems with their child. (page 7, line 230-231)
Comment 3: Also, was there any difference in response from carers who had the spouse/other parent affected by HD still alive?
Answer 3: In paragraph 3.1 this has already been highlighted. Perhaps it is clearer now the following sentence has been added: All parents indicated they thought the child’s behavior was likely caused by the illness of the affected parent, regardless if the parent was still alive or already deceased. However, stepmothers, Francine and Ellen, were convinced it were symptoms and signs of HD. (Page 5, line 166-168).
Comment 4: It's also interesting that if there was a known affected parent, there was still diagnostic delay - is there any literature on this?
Answer 4: We are not aware of any literature on this topic.
Reviewer 2 Report
This is a timely study about the narratives of parents with children suffering from Huntington's disease. Conclusions drawn by the authors are careful and circumstantiated. Of course the number of participants is low, buth this is due to the very low prevalence of this form of the disease. A particular advantaged is the fact that all are from a similar environmental background. However, it would be useful to also have some sense of their cultural background, if at all possible in consideration of personal data protection.
Author Response
Thank you very much for reviewing our paper on juvenile onset of Huntington's Disease. Below you will find the answers to your comments and questions.
This is a timely study about the narratives of parents with children suffering from Huntington's disease. Conclusions drawn by the authors are careful and circumstantiated. Of course the number of participants is low, but this is due to the very low prevalence of this form of the disease.
Comment 1: A particular advantaged is the fact that all are from a similar environmental background. However, it would be useful to also have some sense of their cultural background, if at all possible in consideration of personal data protection.
Answer 1: all interviewed parents or caregivers were from the same cultural background. I can’t give any further information on their ethnical background because of personal data protection. We added the following line: Cultural backgrounds of the parents were alike (page 3, line 118-119).
Reviewer 3 Report
This is a qualitative study evaluating the experience of parents of JOHD subjects. Although written about previously, they report an interesting concept that some parents were actually grateful for a lengthy diagnostic procedure as they were not emotionally ready to face this issue. There are a few suggestions for changes:
Methods: IPA is not defined. It would be helpful to first spell out that acronym and secondly to give a short sentence or two on what this method does.
Results – it is stated that the mean age of diagnosis was 16 with a range of 9 to 24; in the intro, it is stated that the definition of JHD used is diagnosis prior to 21.
Table 1 – There is no age of diagnosis listed for the last entry of Parent Harry
Author Response
Thank you very much for reviewing our paper on juvenile onset of Huntington's Disease. Below you will find the answers to your comments and questions.
This is a qualitative study evaluating the experience of parents of JOHD subjects. Although written about previously, they report an interesting concept that some parents were actually grateful for a lengthy diagnostic procedure as they were not emotionally ready to face this issue. There are a few suggestions for changes:
Comment 1: Methods: IPA is not defined. It would be helpful to first spell out that acronym and secondly to give a short sentence or two on what this method does.
Answer 1:
I added the following line at page 2, line 69-71: IPA’s focus on the in-depth examination of the psychological process and descriptions of how individuals deal with life-transforming, or life-threatening events, conditions or events.
The acronym was already spelled out at page 2, line 68: This study employed in-depth semi-structured interviews and interpretative phenomenological analysis (IPA), a well-established experiential approach in health and clinical psychology
Comment 2: Results – it is stated that the mean age of diagnosis was 16 with a range of 9 to 24; in the intro, it is stated that the definition of JHD used is diagnosis prior to 21.
Answer 2: in the introduction is stated that JHD is defined as HD with an onset < 21 years of age (page 1, line 35-36). As there was a diagnostic delay for most of the patients, the age of diagnosis is somewhat higher than the age of onset. Therefore, the age of diagnosis was >21 years of age for some.
Comment 3: Table 1 – There is no age of diagnosis listed for the last entry of Parent Harry
Answer 3: the reviewer is absolutely right. We apologize for this and I added the age of onset of signs and symptoms in table 1.